Accepted at the ICLR 2024 Workshop on AI4Differential Equations In Science

# Investigating the Effects of Plant Diversity on Soil Thermal Diffusivity using Physics-Informed Neural Networks

**Gideon Stein**[12] **& Sai Karthikeya Vemuri**[1]**,Yuanyuan Huang**[2]**, Anne Ebeling**[3]**,
Nico Eisenhauer**[2]**, Maha Shadaydeh**[1]**, Joachim Denler**[12]
[1] Computer Vision Group, Friedrich Schiller University Jena, Germany
[2] German Centre for Integrative Biodiversity Research (iDiv), Halle-Jena-Leipzig, Germany
[3] Institute of Ecology and Evolution, Friedrich Schiller University Jena, Germany
`{gideon.stein,sai.karthikeya.vemuri}@cuni-jena.de`

## Abstract

The influence of plant diversity on the stability of ecosystems is well-reported in the literature. However, the exact mechanisms responsible for this effect are still a topic of debate. Recently, soil temperature stability was proposed as one possible candidate for such a mechanism. To further evaluate this hypothesis, we investigate the relationship between plant diversity and the thermal diffusivity of the soil during the very dry and hot summer of 2018 in Central Europe. By leveraging Physics-Informed Neural Networks and a 30-minute resolution soil temperature dataset from the Jena Experiment, we find an inverse relationship between plant diversity and the thermal diffusivity of the associated soil. With this, we provide support for the idea of plant diversity as a natural protection against climate-related ecosystem change.

## 1 Introduction

The intricate effects of plant diversity on the stability of ecosystems are well reported in the literature (Isbell et al., 2015), (Loreau et al., 2021), (Tilman et al., 2006). Determining the exact mechanisms that are responsible for these effects is, however, still in progress (De Boeck et al., 2018), (Gross et al., 2014), (Loreau & de Mazancourt, 2013). Recently, (Huang et al., 2024) showed that plant diversity substantially affects soil temperature stability, which in turn influences various mechanisms in the soil. Soil temperature drives a multitude of processes in the ecosystem (Wildung et al., 1975), (Onwuka, 2018), (Guy, 1999), (Nievola et al., 2017). Therefore, stabilizing the abiotic environment of these processes can lead to a generally more stable system. Besides increased biomass, which provides a natural shield against radiation, it was hypothesized that Soil Organic Carbon (SOC), which was reported to be increased through plant diversity (Chen et al., 2020), (Chen et al., 2018), (Spohn et al., 2023) also plays a key role concerning soil temperature stability. SOC is known to influence the thermal diffusivity of the soil (Zhu et al., 2019), (Chen et al., 2012) and, through this, increases soil temperature stability. While this chain is physically well supported, no data-driven studies have investigated this relationship's magnitude.

To close this gap, we perform a data-driven analysis of a multi-depth, long-term, 30-minute resolution soil temperature dataset collected at Europe's largest plant diversity experiment site, the Jena Experiment (Weisser et al., 2017). With this data source, we aim to uncover the relationship between plant diversity and the thermal diffusivity of the corresponding soil. For this, contrary to previous approaches that estimate soil thermal conductivity in a general setting such as (Lukiashchenko & Arkhangelskaya, 2018) or (Arkhangelskaya & Lukyashchenko, 2018), we leverage Physics-Informed Neural Networks (Raissi et al., 2019) (PINNs) to estimate the soil thermal diffusivity constants for selected summer intervals of 2018, a specifically hot and dry year in Central Europe. PINNs provide a versatile framework to learn from data while also incorporating physical knowledge. In this study, we use PINNs in an inverse problem setting to infer the thermal diffusivity of the soil from measurements of temperature at different depths, combined with the partial-differential equation for heat conduction, assuming homogeneous and isotropic conditions. To our

knowledge, this is the first study that directly investigates the relationship between plant diversity and heat diffusivity.

Our results suggest an inverse relationship between plant diversity and the thermal diffusivity of the associated soil during the selected intervals. More specifically, we find that the thermal diffusivity of soil is reduced with increasing plant diversity. With this, we provide data-driven evidence for the stabilizing effects of plant diversity on soil temperature as suggested in (Huang et al., 2024) and support the idea of plant diversity as an effective shield against climate-related ecosystem change.

## 2   DATA

The Jena Experiment (Weisser et al., 2017), situated near Jena, Germany, is a long-term grassland study field site that has operated since 2002. The main site consists of 88 unique plots with different plant diversity levels from 0 (bare ground) to 60 species. The plant diversity level is maintained through frequent weeding year-around. Further, since the field site holds a gradient of soil conditions depending on the distance to the Saale River (east side of the field), the field site is subdivided into four bigger blocks that group the plots according to these conditions. In all plots, soil temperature measurements are taken every minute in depths 5 and 15 cm. Further, in Block 2, additional measurements for the surface temperature and 60 cm depth are taken. We, therefore, focus on data from this block as marked in Figure 1 since every additional measurement can improve the estimation of heat diffusivity. Additionally to these measurements, a weather station is installed in the center of the field site from which we extract the general climatic conditions.

For the scope of this paper, we focus on 2018 (at this time, the hottest year in Central Europe since 1881) and aggregate the data to a 30-minute resolution for each plot individually. To compare the impact of air temperature and general soil moisture availability, we select four three-day intervals (Figure 2) from the summer of 2018 (June to August). Each interval represents some form of extreme weather interval, either having the highest or lowest average air temperature at 2m height or soil moisture at 16cm depth (measured under the climate station). We call these intervals *hot*, *cold*, *wet*, and *dry* correspondingly. Further, we exclude the measurements in 60 cm depth since they show very little variance for the short intervals we selected. While the selected intervals in which we estimate heat diffusivity are sparse so far, we believe they provide a first good insight into the relationship under investigation.

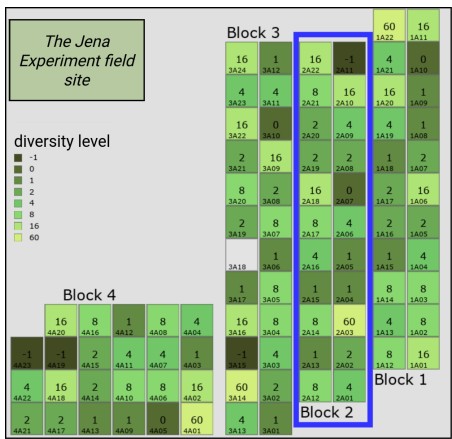
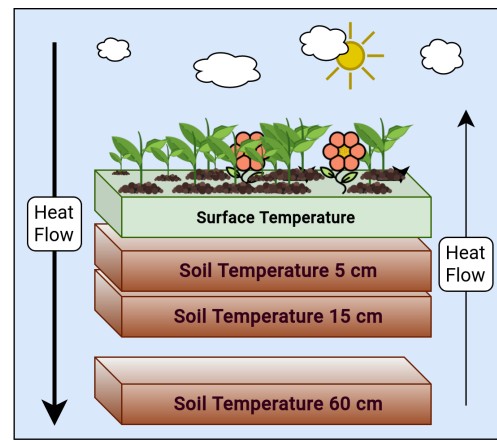

Figure 1: The Jena Experiment field site (left) and the measurements taken in each plot of block 2 (right). Plots with a diversity level of -1 specify uncontrolled plots not considered in this study (together with diversity levels 60 and 0 since these provide only a single sample plot in block 2). In total, we consider 19 plots covering the full diversity gradient.

## 3 METHOD

Heat conduction in solids can be described by a partial differential equation called *heat conduction equation*, which is given as:

$$\frac{\partial T}{\partial t} = K \frac{\partial^2 T}{\partial z^2}, \tag{1}$$

where $T$ is the temperature, $z$ is the spatial coordinate (only depth, in our case since we have a single measurement position per plot), and $t$ is the time index. $K$ is called the *thermal diffusivity* of the soil. This equation is valid for soil, given that the soil is homogeneous and isotropic which we assume here to be approximately true (since we select short intervals that we individually model). While these are arguably strong assumptions, they allow for a first and, more importantly, general picture of the relationship between plant diversity and heat diffusivity since $K$ characterizes the overall thermal properties of soil. As it is a measure of the rate of heat transfer inside the soil, the higher the thermal diffusivity, the faster heat flows through the soil (Carlslaw & Jaeger, 1959).

Physics-Informed Neural Networks(PINNs) (Raissi et al., 2019) offer a paradigm to combine physical knowledge, in the form of differential equations, with deep learning. It involves constructing a multi-objective loss that fuses physics-related and data-related constraints. PINNs can be used to solve both forward and inverse problems involving differential equations. Applications and effectiveness of PINNs for inverse problems are discussed in the literature (Cuomo et al., 2022; Lu et al., 2021; Karniadakis et al., 2021). This study uses PINNs to estimate the soil thermal diffusivity $K$ of a specific plot and a specific interval from multi-depth soil temperature measurements. In other words, we solve the inverse problem involving Equation 1 and the soil temperature data measurements as depicted in Figure 1. Importantly, next to obtaining an estimate for $K$, PINNs also provide an opportunity to interpolate smoothly between depths and time, which is highly beneficial for practitioners as they can match soil temperature measurements with other measurements at the exact same depth and time.

To this end, we construct a feed-forward neural network whose input is the time index $t$ and the depth $z$. Its output is the amplitude and the phase shift for a diurnal sinusoidal cycle of 12 hours that is evaluated at time $t$ to predict the corresponding soil temperature $T$. Consider a neural network that predicts soil temperature, $T(\theta, z, t)$, where $\theta$ denotes the model parameters that must be trained. The corresponding objective of the training is given as:

$$\min_{\theta, K} \underbrace{\left\| \frac{\partial T(\theta, z, t)}{\partial t} - K \frac{\partial^2 T(\theta, z, t)}{\partial z^2} \right\|}_{L_{physics}} + \underbrace{\| T(\theta, z, t) - T_{true}(z, t) \|}_{L_{data}}, \tag{2}$$

where $L_{phyics}$ is the loss term that incorporates physics knowledge (Equation 1) while $L_{data}$ guarantees a proper prediction of data samples. The PINN is trained to optimize the values of $K$ and $\theta$ to find the best representation, satisfying the heat conduction equations and minimizing the error on the data samples. The domain to which the equation and data are restricted is $z \in [0, 15cm]$ and $t \in [0, 3days]$.

We train a PINN for different plots and time intervals, as described in the previous section, to infer the corresponding thermal diffusivities. Concerning hyperparameters, we use AdamW ((Loshchilov & Hutter, 2018)) and train for 200k Epochs, starting with a learning rate of $1e^{-4}$ that is halved after every 50k epochs for all our experiments. We perform full gradient descent (no batching) during all experiments. Further, we use $sin$ as our activation function of choice. Finally, we deploy networks with ten hidden layers and a hidden dimension of 256. We use 432 collocation points (three times as much as the available data points). Finally, since we find that the PINNs do not always converge to a meaningful solution or sometimes cannot produce a reasonably low loss, we, in total, run two independent samples for each data interval and mean the resulting $K$ between all valid runs. We filter runs that result in an MSE loss higher than $1.0$ (loss distribution that we use to determine this threshold is given in Appendix A) and runs that produce values for $K$ that are negative or very close to zero (mathematically possible but physically implausible). We believe that further regularization terms might help restrict the solution space for $K$ to improve consistency even further in the future .

# 4  RESULTS

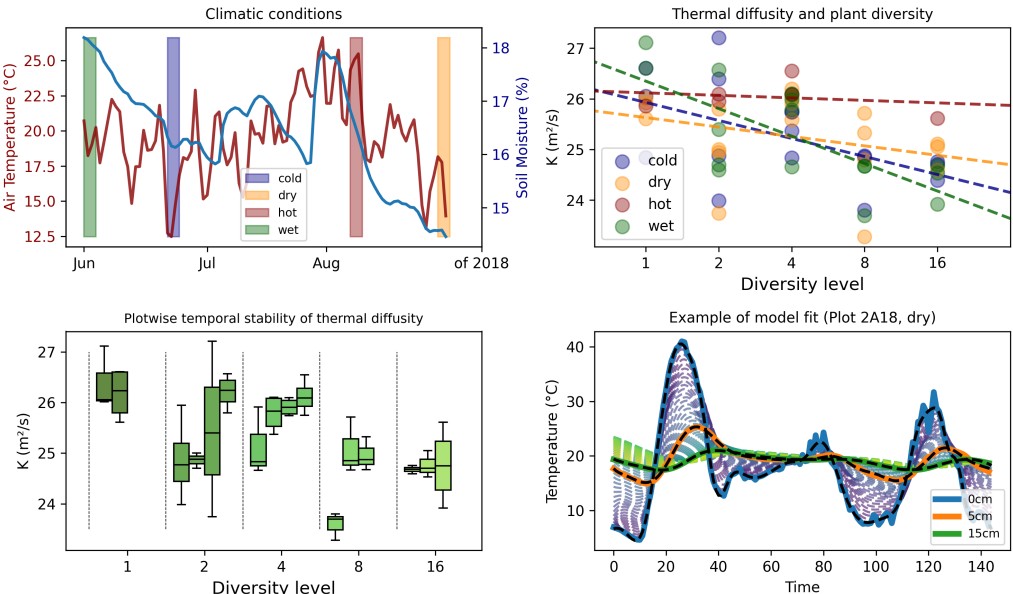

Figure 2: Top-Left: data samples selected to train PINNs Equation 2 independently. Top-Right: The resulting relationship between plant diversity and thermal diffusivity. The thermal diffusivity constant $K$ is generally reduced with diversity. Bottom-Left: The temporal variance of $K$ for individual plots. Bottom-Right: A cherry-picked example of model predictions. The model is able to interpolate smoothly between the data samples (dashed lines). However, it also produces some artifacts close to the 15cm depth.

We report the resulting thermal diffusivity constants and how they relate to plant diversity in Figure 2, alongside the general climatic conditions during the selected intervals and a selected example of a model fit. We found an inverse relation between plant diversity and thermal diffusivity ($p \leq 0.0002$, Figure 2, top-right), tested via a linear mixed effects model with random intercepts, (Oberg & Mahoney, 2007)). These findings are consistent with the results in (Huang et al., 2024) and provide an explanation for the increased soil temperature stability of the soil systems with high plant diversity. Further, we find that the temporal variance of thermal diffusivity of individual plots seems to differ strongly. However, contrary to what we expected based on the results of (Huang et al., 2024), this variance is not generally reduced through plant diversity (Figure 2, bottom-left). Further evidence might provide a clearer picture here as our sample size is limited, and the expected effect might show up only over a longer period of time. Thirdly, we provide predictions of some example PINNs (Plot 2A18 for the *dry* interval) in Figure 2 (bottom-right), to depict the learned representation of the system. We find that the PINN can properly predict the data samples while providing reasonable interpolations between depths (We display all depths from 0-15cm). Finally, since the network architecture we deployed is quite powerful, we evaluated whether our PINNs can find an arbitrary solution to Equation 2 or abuse hidden bias in the data. For this, we performed a permutation test where we shuffled the phase of the Fourier-transformed data and again estimated the thermal diffusivity. The results can be found in the Appendix A. We find that no meaningful patterns can be learned in this setup and there is no relationship between plant diversity and $K$ ($p \leq 0.364$).

# 5  CONCLUSION

In this work, we investigated how plant diversity affects the thermal diffusivity of the corresponding soil, using PINNs. Our first findings show an inverse relationship between plant diversity and thermal diffusivity, confirming this previously hypothesized (most likely indirect, (Huang et al., 2024))

mechanism that governs the stability of soil temperature. These findings support the idea that plant diversity can serve as a natural shield against climate change-related ecosystem stress. Further, we confirm the applicability of PINNs to ecological hypotheses. Here, we have to, however, also note that PINNs have their own specific intricates that have to be kept in mind. Most notably, we found that optimizing for Equation 2 can sometimes lead to models that have low loss but fail to interpolate for all depths inside the domain properly. Such an effect is visible in Figure 2(bottom-right) where temperature interpolations are sometimes outside of the boundary values. We believe that additional regularization might help to overcome this imprecision in the future.

Additionally, while this study provides a first insight into this relationship, it simplifies the soil system, ignoring additional interactions between depth, climatic conditions, soil moisture, phenological cycles, and thermal diffusivity. We plan on investigating these relationships in the future to extend our understanding of the soil system's intricacies. Possible options to incorporate more physical knowledge would be to leverage soil moisture data, use more complex PDEs like the Richardson-Richards equation (Bandai & Ghezzehei, 2022) or heat and moisture transfer models Künzel (1995). Finally, for future work, we plan to extend our work to the full 20 years of data that the Jena experiment provides. With this, we hope to have sufficient data to evaluate how the relationship between thermal diffusivity and plant diversity changes over time and, ultimately, through a broad spectrum of climatic conditions.

## 6 ACKNOWLEDGMENTS

We gratefully recognize the support of iDiv (German Centre of Integrative Biodiversity Research), which is funded by the German Research Foundation (DFG – FZT 118, 202548816). Gideon Stein is funded by the iDiv flexpool (No 06203674-22). Maha Shadaydeh is funded by the Carl Zeiss Foundation within the scope of the program line "Breakthroughs: Exploring Intelligent Systems" for "Digitization—explore the basics (No P2017-01-003), use applications".

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

## A  APPENDIX

Since the threshold that we use to remove non-convergent training runs is chosen by hand, we provide some additional information on the loss distribution of all training runs in Figure 3. With a threshold of one for the MSE loss, we filter less than 10% of training runs.

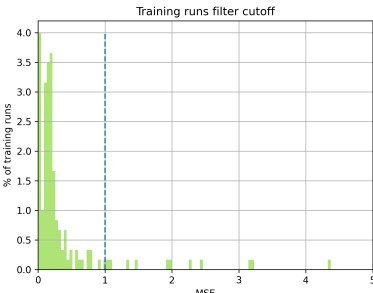

Figure 3: Distribution of the best MSE achieved during training for all PINN runs.

Further, To provide some baseline for significance and to evaluate whether our PINNs somehow leverage hidden bias in the data, we performed a permutation test where we shuffled the phases of the Fourier transformed data and estimated the thermal diffusivity with the exact same setup as for the original data (Figure 4). We find that no meaningful patterns can be learned in this setup and there is no relationship between plant diversity and $K$ ($p \leq 0.364$).

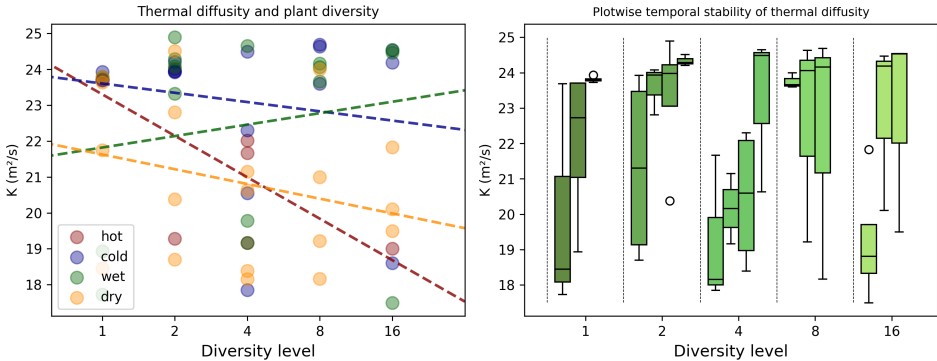

Figure 4: Permutation test results. No significant relationship can be found when fitting PINNs to the permutated data.

