# OpenReview forum: "Investigating the effects of plant diversity on soil thermal diffusivity using Physics- Informed Neural Networks"
_ICLR.cc/2024/Workshop/AI4DiffEqtnsInSci — AI4DiffEqtnsInSci @ ICLR 2024 Poster_

### Official Review · Reviewer_o5Yx · 2024-02-27
**Review by Reviewer o5Yx**

**Rating:** 7
**Confidence:** 2

**Review:**

**Summary.**
The goal of the paper is to study whether plant diversity affects the thermal diffusivity of soil. The existence of this effect is supported by several physical studies. However, the current paper provides the first data-driven study on the magnitude of this effect. The current paper studies the effect using 20 sites from the Jena Experiment dataset. For each site, the thermal diffusivity coefficient is estimated by solving the heat diffusion equation using a physics-informed neural network (PINN). Then, the dependency between plant diversity and the thermal diffusivity coefficient is investigated via a linear mixed effects model with random intercepts. The results indicate a negative correlation between plant diversity and heat diffusivity.

**Discussion.** This paper constitutes a clear and straightforward application of PINNs in science, which closely fits the theme of the workshop. The presentation in the paper is very clear, and the figures are helpful for understanding the setup. The setup of the study is sound, and the paper provides important evidence supporting the existing work on the effects of plant diversity.

In my opinion, it would be helpful to discuss the following aspects in the camera-ready version of the paper.
1. Can other characteristics of the sites provide an explanation for the variability in the thermal diffusivity? For example, it is possible that soil conditions (based on the distance to the river) have an effect on both plant diversity and thermal diffusivity, leading to the observed correlations. How can we rule out this hypothesis?
2. What advantages do PINNs provide over the traditional inverse problem methods for the considered setting?

Minor suggestion: Use `\citep` to put some citations in parentheses.

---

### Official Review · Reviewer_4rcu · 2024-02-28
**Application of inverse PINNs to soil temperature modeling**

**Rating:** 6
**Confidence:** 4

**Review:**

The authors show the application of PINNs to soil temperature modeling in a biodiversity experiment. The study explores the feasibility of  PINNs in estimating the thermal diffusivity of soils under different diversity treatments and 4 different meteorological conditions. Thermal diffusivity is individually estimated per treatment and meteorological condition and subsequently analyzed with mixed-effect modeling. Methodologically, the paper is not too novel, as inverse PINNs have been applied in other domains before. From a domain science standpoint, the paper clearly shows that diversity has an effect on thermal conductivity but does not elucidate how diversity affects thermal conductivity.

Pros:
- application of inverse PINNs to a novel domain that can provide insights into whether PINNs are feasible for noisy ecological data

Cons:
- while the authors clearly state that they are in an exploratory state of their analysis, it falls short in including diversity treatment, soil moisture, or soil organic carbon directly as covariates in their PINNs to disentangle the diversity effect from, for example, an indirect diversity effect via higher evapotranspiration in diverse plots

---

### Meta-Review · Area_Chair_BZ27 · 2024-02-28

**Recommendation:** Accept (Poster)

**Metareview:**

Dear Authors,

Thank you for submitting the draft.

Both reviewers agree that the presented work presents some interesting strengths. However, both reviewers do also raise some points of concern. It is expected that authors will be incorporating comments by the reviewers in the final draft.

regards

AC

---

### Decision · Program_Chairs · 2024-02-29

Accept (Poster)